# Clinical Disease Characteristics and Treatment Trajectories Associated with Mortality among COVID-19 Patients in Punjab, Pakistan

**DOI:** 10.3390/healthcare11081192

**Published:** 2023-04-21

**Authors:** Muhammad Zeeshan Munir, Amer Hayat Khan, Tahir Mehmood Khan

**Affiliations:** 1Discipline of Clinical Pharmacy, School of Pharmaceutical Sciences, Universiti Sains Malaysia, Gelugor 11800, Penang, Malaysia; dramer2006@gmail.com; 2Institute of Pharmaceutical Sciences, University of Veterinary and Animal Sciences, Syed Abdul Qadir Jillani (Out Fall) Road, Lahore 54000, Pakistan; tahir.khan@uvas.edu.pk; 3School of Pharmacy, Monash University Malaysia Sdn Bhd, Jalan Lagoon Selatan, Banday Sunway, Subang Jaya 45700, Selangor, Malaysia

**Keywords:** COVID-19, SARS-CoV-2, mortality, comorbidity, pharmacological therapies, biomarkers, outcomes

## Abstract

Background: Data on Pakistani COVID-19 patient mortality predictors is limited. It is essential to comprehend the relationship between disease characteristics, medications used, and mortality for better patient outcomes. Methods: The medical records of confirmed cases in the Lahore and Sargodha districts were examined using a two-stage cluster sampling from March 2021 to March 2022. Demographics, signs and symptoms, laboratory findings, and pharmacological medications as mortality indicators were noted and analyzed. Results: A total of 288 deaths occurred out of the 1000 cases. Death rates were higher for males and people over 40. Most of those who were mechanically ventilated perished (OR: 124.2). Dyspnea, fever, and cough were common symptoms, with a significant association amid SpO2 < 95% (OR: 3.2), RR > 20 breaths/min (OR: 2.5), and mortality. Patients with renal (OR: 2.3) or liver failure (OR: 1.5) were at risk. Raised C-reactive protein (OR: 2.9) and D-dimer levels were the indicators of mortality (OR: 1.6). The most prescribed drugs were antibiotics, (77.9%), corticosteroids (54.8%), anticoagulants (34%), tocilizumab (20.3%), and ivermectin (9.2%). Conclusions: Older males having breathing difficulties or signs of organ failure with raised C-reactive protein or D-dimer levels had high mortality. Antivirals, corticosteroids, tocilizumab, and ivermectin had better outcomes; antivirals were associated with lower mortality risk.

## 1. Introduction

Since the outbreak of the COVID-19 pandemic, it has become increasingly evident that certain populations are at higher risk for worse disease outcomes. These include the elderly, individuals with pre-existing medical conditions, and those with weakened immune systems [1]. The progression of COVID-19 can vary widely depending on the presence of the risk factors. While some individuals may experience only mild symptoms, others may rapidly progress to severe disease and require hospitalization with advanced medical interventions. Previous literature has reported that 14–23% of those hospitalized, develop a serious and potentially fatal condition [2,3]. Understanding the association of risk factors with COVID-19 disease progression is critical in developing effective patient care plans for those at risk of dying due to COVID-19 infection [4].

Age is one of the most significant demographic predictors of poor disease progression in COVID-19. Older patients, particularly those over 65 years of age, are at higher risk for severe disease, hospitalization, and death. A study conducted in Wuhan, China, found that patients over the age of 65 had a significantly higher risk of mortality than younger patients. Similarly, a study conducted in the United States found that the risk of hospitalization and death increased with age [5,6].

Comorbidities, such as hypertension, diabetes, cardiovascular disease, and chronic respiratory conditions, have also been identified as predictors for poor disease progression in COVID-19. A systematic review and meta-analysis of 30 studies found that patients with comorbidities were more likely to require hospitalization, admission to the ICU, and mechanical ventilation. Another study found that the presence of two or more comorbidities was associated with a higher chance of mortality in COVID-19 patients [7,8]. Diabetic individuals are more likely to acquire disease due to increased cellular binding and entry affinities, decreased viral clearance, diminished T-cell activity, concurrent heart disease, and hyper-inflammation due to cytokine storms [9]. Similarly, the new coronavirus infection can cause myocardial injury, dysrhythmias, acute coronary syndrome (ACS), venous thromboembolism, and coronary artery disease. People who have a history of cardiovascular disease or who have an overabundance of cardiovascular risk factors are also more prone to viral infections and have a lower chance of surviving [10,11].

Laboratory findings, such as elevated levels of D-dimer, lactate dehydrogenase (LDH), and C-reactive protein (CRP), have also been associated with poor disease progression in COVID-19 [12]. Additionally, the potential role of Adiponectin (APN), which is an adipocyte primarily produced by visceral adipose tissue in COVID-19 has been described by various studies [13]. It plays both an anti-inflammatory and pro-inflammatory role and can prevent SARS-CoV-2-induced acute lung injury [14]. Hyper-inflammation-induced adipose tissue dysfunction has been found to reduce APN production in patients with severe COVID-19 and respiratory failure [15,16]. On the other hand, a study conducted in Italy found that patients with higher levels of D-dimer and LDH were also more likely to require admission to the ICU and mechanical ventilation. Similarly, a study conducted in China found that patients with elevated CRP levels had a higher risk of mortality [17,18].

Radiographic findings, such as the presence of ground-glass opacities and consolidations on chest CT scans, have also been associated with poor disease progression in COVID-19. A study conducted in Wuhan, China, found that patients with severe lung involvement on chest CT scans had a higher risk of mortality than those with milder involvement. Another study found that the extent of lung involvement on chest CT scans was positively correlated with the severity of illness in COVID-19 patients [19,20].

It is noteworthy, that the mortality rate among COVID-19 patients can also vary depending on other factors, including geographic location, access to healthcare, and the overall socioeconomic status of the country [21]. Despite harsh measures including travel bans, social isolation, lockdowns, and heightened testing, many countries have struggled to contain the spread and reduce the death toll. Surprisingly, lesser socioeconomic development and social overcrowding are not linked to increased death rates. One of the primary worries has been the impact of this epidemic on poorer nations with inadequate infrastructure and healthcare systems [8,22]. According to data from the World Health Organization (WHO), as of September 2021, there have been over 1.2 million confirmed cases of COVID-19 in Pakistan, with over 27,000 deaths. The number of cases peaked in June 2021, with over 6000 new cases reported daily. However, since then, the number of cases has decreased significantly, with an average of around 2000 new cases reported daily in September 2021 [23].

Treatment trajectories can also affect the mortality rate of COVID-19 patients. Early detection and prompt treatment can help to reduce the severity of the disease and prevent complications that may lead to death. Clinical studies and medication repurposing have helped to identify a number of effective treatment plans. Some of the treatment options include antiviral medications, immunomodulatory drugs, oxygen therapy, mechanical ventilation, and extracorporeal membrane oxygenation (ECMO). The choice of treatment will depend on the severity of the illness and the individual patient’s characteristics, such as age, comorbidities, and immune status [24].

Antiviral drugs such as Remdesivir have been shown to reduce the length of hospital stays and improve recovery time in hospitalized COVID-19 patients [25]. Another antiviral drug, Molnupiravir, has shown promising results in clinical trials and is currently under emergency use authorization by the FDA in the US [26]. Immune modulators such as corticosteroids including dexamethasone have also been shown to reduce mortality rates in critically ill COVID-19 patients [27]. Other immune modulators such as tocilizumab and barcitinib have also been used to treat severe COVID-19 cases. Monoclonal antibodies such as casirivimab/imdevimab and sotrovimab have been authorized by the FDA for the treatment of COVID-19 in certain patient populations. Oxygen therapy is often used to support patients with respiratory distress. This can include supplemental oxygen through a nasal cannula, non-invasive ventilation, or mechanical ventilation in severe cases [28]. Convalescent plasma therapy involves transfusing blood plasma from recovered COVID-19 patients to those who are currently infected. This therapy has shown promise in reducing the severity of illness in some COVID-19 patients [29].

This research will investigate numerous factors linked with COVID-19 mortality in high-risk groups and assess illness development in these individuals to help enhance current and future pandemic management. We can create more effective preventative and treatment techniques to lessen the effects of this pandemic on vulnerable groups by better understanding these variables. It is also becoming increasingly clear that different health management decisions made by the healthcare systems around the world over the pandemics multiple cycles had a significant influence. Therefore, due to the scale and complexity of COVID-19 management, it is critical to construct organized care in order to avoid extra illogical and inconsistent patient management.

## 2. Materials and Methods

### 2.1. Study Design

This multi-institution retrospective observational study was conducted from March 2021 to March 2022, after the approval of the Institutional Review Board of the University of Veterinary and Animal Sciences, Lahore (IPS-IRC96/02/2021), and with the approval of participating hospitals, using the health record data collected prospectively.

### 2.2. Setting and Sampling

The study was conducted in all public-sector hospitals of Lahore and Sargodha, two major cities of Punjab selected using a two-stage cluster sampling technique. The combined population of these two cities is roughly 14 million. The medical record of COVID-19 patients was extracted after the approval of the participating institutes. With 90% power of the study and 5% level of significance, and an effect size of 0.11, a sample size of 871 participants was calculated, which was raised to 1000. The systematic sampling technique was employed within the clusters, and a record of every fifth patient was included in the study [30].

### 2.3. Data Collection

Data were manually extracted from patient medical health records using a specially designed data collection form. Patient’s sociodemographic characteristics, underlying comorbidities, initial clinical presentation, baseline vital signs, laboratory data, and pharmacological medications are given, and patient discharge and survival status were recorded. Relevant clinical data from 1000 cases were reviewed and tracked until the treatment outcome of either death or cure was identified.

Laboratory investigation which was recorded were total blood count, levels of C-reactive protein (CRP), and D-dimer. An increase in liver enzymes and serum creatinine from baseline was regarded as acute liver and kidney damage.

Pharmacological interventions involving drug classes such as antibiotics, antivirals, corticosteroids, monoclonal antibodies, anticoagulants, and ivermectin, were recorded as an important part of the study data.

### 2.4. Inclusion and Exclusion Criteria

Patients with complete file records, who were COVID-19 positive and confirmed by the PCR testing were included in the study. Incomplete files were discarded, patients who died within the 1st day of hospital admission, suspected COVID-19 patients who were not confirmed by PCR testing, and patients given non-pharmacological supportive treatment or not given any pharmacological treatment were not included.

### 2.5. Outcome Variables

We noted the COVID-19 infection occurrence, intensive care admission, and mechanical ventilation during the different pandemic waves in Pakistan. However, treatment outcome such as mortality was the only primary independent outcome variable correlated with the dependent variables of the studied patients.

### 2.6. Statistical Analysis

Data obtained were subjected to descriptive analysis using statistical package for social science (SPSS) software version 20.0 (SPSS, Inc., Chicago, IL, USA). Frequencies and percentages are used to display categorical variables. Continuous variables, on the other hand, were expressed as mean. All other measures were preserved as independent variables while the clinical outcomes (cured/death) were kept as dependent variables. The Pearson chi-square test was used to examine the correlation between the variables and primary outcomes. Furthermore, logistic regression models were used to determine the association between the dependent variable primarily mortality, and any of the independent variables with a specific focus on clinical disease characteristics and pharmacological drugs administered. Odds ratios and their accompanying confidence intervals were used to evaluate this. These univariate characteristics were then subjected to multivariate analysis to determine the predicted parameters for COVID-19 outcomes in terms of mortality. *p*-value < 0.05 was used to identify statistically significant observations throughout the statistical analysis.

## 3. Results

Table 1 summarizes the sociodemographic profile and clinical characteristics of the patients studied. There were a total of 1000 Pakistani patients included in the study. The research population was 51 years of age on average, with the highest rates of cases (72.7%) distributed among those aged ≥ 40 years. Deaths were much higher among older vs younger participants (mean age 55 years vs. 49 years). There was a male preponderance, with slightly more than half (59.7%) of all patients being male. The death rate was lower in females than in males (34.3% versus 65.6%). The average length of stay in the hospital or time to clinical remission was 8.9 days. There were 288 deaths among these individuals, with an estimated average mortality rate of 28.8%. Any comorbidity was present in 52.1% of survivors (compared to 69.4% of nonsurvivors), with hypertension being the most common chronic condition, followed by diabetes and ischemic heart disease. Furthermore, the nonsurvivors had a higher prevalence of hypertension, diabetes, and ischemic heart disease (54.1%, 48.6%, and 16.6%, respectively).

The patients were also evaluated for their clinical symptoms and vital signs. According to the findings in patients with SARS-CoV-2, the body develops a fever, upper respiratory symptoms, and ultimately symptoms of the lower gastrointestinal (GI) tract. Fever and respiratory tract symptoms such as dry cough and dyspnea were found to be among the most common. Dyspnea was recorded in 79.5% (n = 795), fever in up to 63.7% (n = 637), and cough in 40.9% (n = 409). Table 2 summarizes the symptoms that were present at the time of diagnosis. Dyspnea was found in 91% of nonsurvivors, fever in 60%, and cough in 41.6% of those who died. Vital signs were inconsistent, with a mean body temperature and heart rate of 99.3 °F and 96.5 beats/minute, respectively. Almost half of the evaluated cases had some degree of respiratory failure. The oxygen saturation was found to be lower in 75% of the total cases, with an average of 88.6%. Furthermore, 79.5% had a high respiratory rate, with a mean of 24 breaths/minute. Mean systolic and diastolic blood pressures were 139.4 mmHg and 88.8 mmHg, respectively. One-quarter of the patients required mechanical ventilation, and 74.3% (214/288) of the patients who died were on ventilation.

Patients had pathological changes during COVID-19, as confirmed by various laboratory tests. Table 3 illustrates these variations. During the course of the infection, a high white blood count was found in 69.1% of the patients, with a mean value of 15.47 × 109/L. Lymphocytopenia was detected in 72.5% of the patients, with a mean value of 14.8%. Variations in liver and kidney function parameters were also identified following the infection. Liver enzyme levels were determined to be above the normal reference range, with mean values of AST = 61.76 units/L and ALT = 81.4 units/L. Renal impairment was also evident, as seen by high urea (94%) and serum creatinine (24.7%) levels in the studied patients. In terms of inflammatory indicators, C-reactive protein (CRP) and D dimer levels were both elevated above the normal range. D-dimer levels were beyond the upper limit in 77.0% (222/288) of patients who died, with a mean value of 0.935, whereas 624/1000 cases had CRP levels above the normal concentration, with a mean of 53.32 mg/L, demonstrating a clear link with mortality.

Table 4 illustrates the odds ratios calculated using the logistic regression model. It compiles research on risk variables linked to death. In univariate analysis, all demographic factors, clinical parameters, and laboratory findings were recognized as independent predictable variables with statistical significance between outcomes such as cure and death. Age ≥ 40 years (OR: 1.925, *p* < 0.001 *), male sex (OR: 1.422, *p* < 0.015), and having any comorbidity (OR: 2.089, *p* < 0.001 *) were independently associated with death outcomes. Patients who had tachycardia (OR; 1.736, *p* < 0.001 *) or elevated blood pressure (OR: 2.021, *p* < 0.001 *) also had higher odds of mortality.

In laboratory feature analysis, patients with clinical deterioration reported the development of liver and kidney dysfunction indicated by elevated levels of liver enzymes such as AST (OR: 2.55, *p* < 0.0001 *), serum creatinine (OR: 2.359, *p* < 0.001 *), and hyper-inflammation or coagulation markers such as CRP (OR: 3.790, *p* < 0.001 *) and D dimers (OR: 4.239, *p* < 0.001 *). Death related to COVID-19 disease was also associated with hyperglycemia, lymphocytopenia, and thrombocytopenia.

The univariate analysis yielded 23 independent variables, which were used in the multivariate analysis to find reliable predictors of COVID-19-related mortality. Patients on a ventilator (OR = 124.26, *p* < 0.001 *) with symptoms of respiratory distress (SOB, OR: 2.595, *p* = 0.015 *and low oxygen saturation OR: 3.234, *p* = 0.002 *) along with elevated levels of D dimers (OR: 1.659, *p* = 0.046 *) and CRP (OR: 2.979, *p* ≤ 0.001 *) and the development of acute liver or kidney disease indicated by elevated levels of bilirubin (OR: 2.409, *p* = 0.038 *) and serum creatinine (OR: 2.344, *p* = 0.002 *) had significantly greater odds of death from COVID-19 infection.

The drugs most commonly used to treat COVID-19 patients were antibiotics (n = 779; 77.9%), particularly azithromycin. Corticosteroids (n = 548; 54.8%), ivermectin (n = 92; 9.2%), and tocilizumab (n = 203; 20.3%) were also prescribed. Anticoagulants were utilized in 34.0% (n = 340) of instances, with enoxaparin and unfractionated heparin being the most commonly used In terms of medications administered and treatment outcomes, patients treated with antivirals (OR: 0.475, *p* = 0.038 *), corticosteroids (OR: 0.644, *p* = 0.189), tocilizumab (OR: 0.896, *p* = 0.748), and ivermectin (OR: 0.518, *p* = 0.129) had a higher survival chance; however, only antivirals were significantly associated with lower death odds compared to other medications. On the other hand, antibiotics (OR: 1.316, *p* = 0.475) and anticoagulants (OR: 1.378, *p* = 0.254) were not positively associated with improved clinical outcomes in the studied COVID-19 patients.

## 4. Discussion

A retrospective study of 1000 infected individuals in Pakistan confirmed by PCR test findings is provided to investigate determinants of mortality owing to COVID-19 infection. We looked at the patient’s demographics, comorbidities, clinical symptoms, laboratory findings, and pharmacological therapy in relation to clinical outcomes that is mortality rates because identifying risk factors is crucial for effective health efforts to combat this disease. The majority of the report’s findings on sociodemographic data, comorbidities, and pharmacological treatment were consistent with comparable studies conducted around the world [31,32].

The nonsurvivor group was significantly older than the survivor group and had more comorbidities, which is consistent with earlier research. Our study’s average age was 51 years, which was similar to research conducted in the United States [33]. This could be because older patients have higher host innate responses to the virus, and abnormalities in T and B cell activity could also result in a deficiency in viral replication control, leading to patient death [34,35]. In terms of gender, our findings show a higher proportion of males, as previously documented by other research, and that the majority of cases who died were likewise men [36,37]. This gender component, as well as greater rates in males, may be related to a general demographic fact of lower life expectancy in men in general across the world [38].

This study’s fatality rate is 28.8%, and the estimated mortality rate in Italy was 12.7%, which is significantly higher than the number recorded in South Korea (approximately 2%) and even death rates in different parts of Italy vary greatly. Lazio (Central Italy) has 5.7%, whereas Northern Italy (Lombardy) has 18.4% [39,40]. This may be due in part to several variables such as political management of the health crisis, availability of ventilators, and demographic considerations (e.g., the differing median age of patients) [41,42].

In this report, 57% of the observed cases had one or more underlying health conditions, with hypertension described as the most common, followed by diabetes and cardiovascular disease. Diabetes has been identified as a risk factor for severe infection and even death [43]. Although the exact cause of the poor prognosis of diabetic patients is unknown, various factors may play a role. For example, patients with COVID-19 are more likely to have poorly managed diabetes, which compromises the immune response. Second, since ACE inhibitors are used in many diabetic patients, ACE 2 receptor inhibition results in reduced insulin sensitivity and can promote viral entry into host cells. Third, pancreatic cell function can worsen in some COVID-19 individuals when the virus causes damage to the pancreatic islets, resulting in new-onset acute diabetes [43,44,45]. Another pre-existing ailment, such as cardiovascular problems, has also been considered a risk factor. This connection is unclear, and numerous mechanisms have been postulated. One possibility is that the virus infects cardiomyocytes directly, as because the pulmonary circulation’s initial objective is the heart, the virus could infect vascular cells in the pulmonary artery, attract immune cells and trigger an inflammatory reaction such as a cytokine storm [46,47]. Systolic blood pressure > 140 mmHg has also been found to be a significant predictor, with hypertension found to be more frequent among nonsurvivors. Initially, hypertension was reported in a case series from China and Europe in 27% to 30% of the patients and was associated with increased COVID-19-related deaths [48].

Patients admitted to the ICU, predictably, required more intrusive mechanical ventilation, and their mortality rate was much greater. 90.2% of the COVID-19 patients in the nonsurvivor group were on invasive ventilation. The presenting symptoms were varied, but the most common symptoms reported on admission (dyspnea, fever, and cough) were consistent with previous research. Several studies have highlighted the possible impact of shortness of breath on patient outcomes. Persistent dyspnea in COVID-19 patients should raise an alarm since it indicates clinical worsening and poor lung compliance. Low oxygen saturation was also a significant predictor of mortality [49].

A substantial number of patients showed abnormal laboratory values, as indicated in all prior series. Increased blood levels of CRP, BUN, and SrCr and a higher D-dimer level all exhibited an independently increased risk of poor disease progression and mortality, similar to findings in the United States [50,51]. Hyperglycemia was also found in high numbers and many observational studies have discovered that not only diabetes but also hyperglycemia during COVID-19 infection was linked to poor outcomes and death [52,53]. Another laboratory anomaly seen in COVID-19 patients is lymphopenia. This implies immune system dysfunction. Thrombocytopenia with elevated D-dimer levels is also prevalent in COVID-19 infection and linked to unfavorable outcomes. D-dimer elevation indicates that patients are more likely to have venous thromboembolism. Those who died had almost 3-fold increased levels of D-dimer, which is similar to an observation from another such analysis [54]. Even after multivariate correction, higher levels of D-dimer were also strongly linked with mortality in other retrospective analyses. This could be attributable to direct viral endothelial damage as well as sepsis-induced coagulopathy in some patients [55]. Inflammatory biomarkers such as CRP exhibited significantly higher values in multivariate analysis as a predictor of mortality. In a comparable study in New York City, nearly all patients with an elevated CRP value demonstrated signs of a systemic inflammatory response to SARS-CoV-2 infection. Recent studies have also found a link between high CRP levels and respiratory failure, with a higher risk of AKI, VTE, and ARDS necessitating mechanical ventilation [56,57]. SARS-CoV-2 has also been found to damage other organs in the body, such as the kidneys and liver, and may also exacerbate the existing chronic disease of these organs. In this study, elevated AST was found in a major percentage of the total deceased cases. According to a previously published report on the same subject, several individuals with COVID-19 suffered acute liver injury during the course of their disease, with most of them dying [58]. The damage could be linked to the fact that the liver is a possible target for the coronavirus due to the increased expression of ACE2 receptors in hepatocytes or it can be produced directly by medication hepatotoxicity [59]. Acute renal damage markers, such as serum creatinine levels, have also been demonstrated to be independent predictors of death. AKI was found in approximately 24.7% of COVID-19 patients. Several investigations from the United States and Italy also found similar results. Possible explanations are the activation of cytokine storms linked to renal tissue destruction [60,61].

The patterns of use of drugs recommended for pharmacological interventions vary based on geography. Research from other countries has shown that azithromycin is widely used (60.6–88.6%), which is consistent with our findings. Similarly, the majority of patients got some systemic corticosteroids in contrast to others who were prescribed fewer [62,63,64]. Tocilizumab is another medication that has accumulating evidence in its favor. In this report, it was prescribed to a small percentage of people, compared to earlier studies [65]. In a univariate assessment of medications administered, we found no evidence that the use of treatments such as antibiotics, corticosteroids, anticoagulants, antivirals, or even tocilizumab reduces mortality in COVID-19 patients. However, multivariate analysis revealed that corticosteroids, antiviral medications, tocilizumab, and ivermectin were associated with lower probabilities of mortality.

A meta-analysis of all types of clinical studies (including 17 papers) found no clinical improvement in COVID-19 patients given azithromycin in terms of mortality, mechanical ventilation, or hospitalization [66]. In a review by Langford et al., antibiotics were given to 74.5% of patients, although bacterial coinfection was estimated to be 8.6% [67]. In the absence of substantial scientific evidence, preliminary information about COVID-19 treatments, combined with a willingness to try what is available, may have influenced antimicrobial prescribing habits [68]. However, during the challenging times of the pandemic, antibiotic stewardship principles must not be ignored to protect the efficacy of current antimicrobials such as azithromycin.

Corticosteroids have been shown to be beneficial in several observational studies of COVID-19 interstitial lung disease, which is consistent with our findings [69]. It has been found, however, that patients receiving corticosteroid therapy may develop complications such as ARDS, sepsis, AKI, and multiorgan failure. A Spanish investigation also discovered that corticosteroid-induced immunosuppression can impair pathogen clearance and increase the risk of death [70]. These findings may come as a surprise, given that steroids were formerly the most effective treatment for this disease. Some doctors believe that glucocorticoids should only be taken when absolutely necessary. Dexamethasone, for example, should only be given to patients who require ‘increasing doses of supplemental oxygen’, depending on their clinical condition [71]. Systemic corticosteroids should be tapered off as soon as possible in relation to the clinical response [72]. In the survival analysis, patients in the low-dose corticosteroid group had a significantly better likelihood of survival, and hyperglycemia or leukocytosis was more common in the high-dose group [73,74]. Therefore, Low-dose corticosteroid therapy is the standard of care for COVID-19 hospitalized patients who require supplemental oxygen [75,76].

Anticoagulants were also more common among non-survivors. Certain retrospective observational evidence suggests that anticoagulation above preventative doses is associated with an increased risk of bleeding [77]. One study discovered that patients who had corticosteroid treatment in conjunction with an anticoagulant had a greater risk of death, as compared to those who did not receive steroids [78]. Despite this, anticoagulants are an important supportive treatment that may increase patient survival. LMWH is found to be associated with lower mortality in severe patients with elevated D-dimer levels [79]. Current guidelines do not recommend that persons hospitalized with COVID-19 receive more than preventive doses of anticoagulation [80]. The timing of anticoagulant drug initiation may also be essential in the therapy of COVID-19 patients. There is just a little window of time to commence treatment and effectively prevent clinical deterioration [81]. As a result, higher-dose anticoagulation may be considered, if at all, in noncritically ill COVID-19 patients hospitalized, whose early disease condition may still give therapeutic benefit [82]. ISTH guidelines for antithrombotic treatment in COVID-19 patients hospitalized with non-critically ill COVID-19 highly advocated taking a prophylactic dosage of low molecular weight heparin or unfractionated heparin (LMWH/UFH) [83].

The study findings also provide strong support for early antiviral medication as a primary treatment strategy. Remdesivir has been related to a decreased risk of death, implying that starting antiviral treatment during the earliest stage of sickness may give a therapeutic benefit. In a meta-analysis of 5 placebo-controlled clinical studies including a total of 13,594 patients, those given remdesivir had a faster clinical recovery, although the differences in mortality were not statistically significant [84]. In Spain, an observational study was carried out. The use of remdesivir improved survival in patients who received medication earlier, with a total risk of mortality reduction of 62%. International remdesivir trials in Japan also revealed that an antiviral medicine reduces the probability of hospitalization [85,86]. Remdesivir is thus recognized in guidelines as an effective medicine. Similarly, the IDSA recommends remdesivir for hospitalized patients but not for critically ill patients [87].

Tocilizumab was also reported to lower mortality in persons with inflammation (CRP > 75 mg/L) in some observational studies [88]. Viral infection lowers the synthesis of interleukin (IL)-17, which is required for neutrophils to remove germs. Furthermore, influenza virus infections encourage S. aureus colonization [89]. Several authors discovered that tocilizumab therapy significantly reduced CRP and D dimer levels. As a result, early tocilizumab administration is recommended in these patients for a higher survival rate [90]. Increased levels of some plasma indicators, such as CRP, may aid in the decision-making process for deciding which patients will benefit from tocilizumab. Study shows that tocilizumab treatment had a therapeutic impact on people with moderate-to-severe COVID-19 in the early phases of clinical deterioration, especially when paired with corticosteroid therapy [91].

Ivermectin use was also linked to lower risks of infection, hospitalization, and death. Another study compared two doses of ivermectin to standard therapy alone. Ivermectin had a lower death rate (15.0% vs. 25.2%; *p* = 0.03). This medicine has been proven to significantly lower vital load, reduce the time to negative RT-PCR test results, and enhance. These findings were more pronounced in severe individuals. However, the evidence for ivermectin role in COVID-19 infection is conflicting. At least three meta-analyses of ivermectin trials discovered a significant therapeutic advantage, whereas others did not [92,93].

COVID-19 mortality varies greatly between high- and low/middle-income countries. The developed world has the most information on this subject. At the same time, data from developing countries still need to be included, particularly in the South Asian region, home to one-fifth of the world’s population. Aside from the clinical features of the patients, there are several surprising correlations between COVID-19-related mortality and varying patterns of population densities as well as socioeconomic and environmental factors between regions. Another unexpected finding is that COVID-19 fatality rates are more significant in high-income countries than in low- and low-middle-income countries, despite developed countries having superior access to healthcare, hygiene, and sanitation. This study could also look into this element of the COVID-19 pandemic.

Therefore, we present a COVID-19 mortality forecast study based on many patients in a middle-income and densely populated country such as Pakistan, which shares borders with China, Iran, and India and has related socioeconomic and demographic traits. Numerous initiatives have been launched on the same subject. However, most of the proof is founded on research done in the early phases of the pandemic. The continued investigation and accumulation of data from each period will also help shed light on critical lessons for future generations.

This will also assist to determine the treatment plan and priority group for immunization or medical care and should be publicized in the country’s guidelines. Furthermore, COVID-19 treatments have evolved, and the value of current pharmacological regimens is still up for discussion. This research can also assist medical professionals in identifying high-risk patients who will require specialized treatment quickly and developing emergency triage strategies. Such patients are more likely to become severely sick and need admittance to the ICU; by doing this, more ICU spaces can be spared. These epidemiological studies are also essential to support logistical preparation for enhancing patient treatment and the caliber of resources strained healthcare systems to deal with such pandemics.

## 5. Conclusions

The COVID-19 epidemic has had a significant effect on social, economic, and health conditions, pushing healthcare systems to their breaking point all over the world. According to this research, there were recurring risk factors for rising mortality across the pandemic phases. Age, male sex, and pre-existing comorbidities are revealed to be demographic risk factors for the onset of severe illness and even mortality. The intensity and mortality of COVID-19 may also be predicted by laboratory markers that point to problems during hospitalization, such as the onset of acute respiratory distress syndrome or multiple organ failure. Therefore, COVID-19 patients who have mortality indicators should be quickly admitted to the hospital, evaluated, and treated before their clinical state deteriorates. Additionally, in order to protect global health from the current COVID-19 pandemic and future dangers, it is essential to guarantee health equity in less developed nations with subpar health services to develop systems that allow for early access to healthcare, which are essential for lowering the mortality rate from severe COVID-19.

Strengths of our multicenter report are the relatively large number of patients from two geographically diverse hospitals in Pakistan, as well as the real-world settings and use of simple clinical parameters as a tool to predict COVID-19 mortality; other strengths are its applicability in triaging patients at the time of admission and use of logistic regression model in order to examine the association among the variables and COVID-19 related mortality.

There are also evident limitations. In this retrospective analysis, there is no control group. The therapies were administered in a range of doses, modes, and durations, and the severity of COVID-19 disease was not consistent, ranging from patients who did not require oxygen to mechanically ventilated patients in intensive care units. The patients were recruited from a specific region, and our results might not apply in other countries as factors associated with mortality may differ in various regions.

## Figures and Tables

**Table 1 healthcare-11-01192-t001:** Patient-related information with confirmed diagnosis of SARS-CoV-2 Infection (n = 1000).

Variables	Groups	N %	Cured N (%)	Death N (%)	*p* Value
Age Mean age: 51.17	Below 40 years	273 (27.3)	219 (80.2)	54 (19.7)	<0.001 *
	Above 40 years	727 (72.7)	493 (67.8)	234 (32.1)	
Gender	Male	597 (59.7)	408 (68.3)	189 (31.6)	0.015 *
	Female	403 (40.3)	304 (75.4)	99 (24.5)	
No. of days to recovery Mean: 8.9	1–15 day	870 (87)			
	15–30 days	125 (12.5)			
	More than 30 days	5 (0.5)			
Treatment Outcome	Cured	712 (71.2)			
	Death	288 (28.8)			
Any co-morbidities	Yes	571 (57.1)	371 (37.1)	200 (20.0)	<0.001 *
Diabetes	Yes	416 (41.6)	276 (66.3)	140 (33.6)	0.004 *
Hypertension	Yes	444 (44.4)	288 (64.8)	156 (35.1)	<0.001 *
IHD	Yes	147 (14.7)	99 (67.3)	48 (32.6)	0.264

* Chi square; * *p*-value < 0.05 is statistically significant.

**Table 2 healthcare-11-01192-t002:** Clinical characteristics on initial presentation.

Variables	Baseline Value	Mean	N (%)	Cured N (%)	Death N (%)	*p*-Value
Fever	More than 98.6 °F	99.3 °F	637 (63.7)	463 (72.6)	174 (60.1)	0.170
Tachycardia	More than 100 Beats/minute	96.5 Beats/minute	306 (30.6)	193 (27.1)	113 (39.2)	<0.001 *
Respiratory Rate	More than 20 Breaths/minute	24.1 Breaths/minute	795 (79.5)	532 (74.7)	263 (91.3)	<0.001 *
Oxygen Sat %	Less than 95%	88.6%	750 (75.0)	504 (70.7)	246 (85.4)	<0.001 *
Cough	Yes		409 (40.9)	289 (40.5)	120 (41.6)	0.754
Blood pressure	Above 140/80 mmHg		399 (39.9)	249 (34.9)	150 (52.0)	<0.001 *
Systolic: Diastolic:		139.4 mmHg 88.8 mmHg				
On ventilator	Yes		237 (23.7)	23 (3.2)	214 (74.3)	<0.001 *

* Chi square; * *p*-value < 0.05 is statistically significant.

**Table 3 healthcare-11-01192-t003:** Laboratory parameters of patients having SARS-CoV-2 Infection.

Variables	Groups	Baseline Value	N (%)	Mean	Cured N (%)	Death N (%)	*p*-Value
CBC	WBC	>10 × 10^9^/L	691 (69.1)	15.47	455 (65.8)	236 (34.1)	<0.001 *
	Lymphocyte %	<40%	725 (72.5)	14.88	503 (69.3)	222 (30.6)	0.039 *
	Platelet count	<150	171 (17.1)	260.71	104 (60.8)	67 (39.1)	0.002 *
LFT	AST	>40 units/L	437 (43.7)	61.76	264 (60.4)	173 (39.5)	<0.001 *
	ALT	>56 unites/L	400 (40.0)	81.45	273 (68.2)	127 (31.7)	0.093
	Bilirubin	>1.2 mg/dL	75 (7.5)	0.922	27 (36)	48 (64)	<0.001 *
RFT	Creatinine	>1.2 mg/dL	247 (24.7)	1.462	141 (57.0)	106 (42.9)	<0.001 *
	Urea	>20 mg/dL	940 (94.0)	55.51	666 (70.8)	274 (29.1)	0.335
Others	D-dimers	>0.50	537 (53.7)	0.935	315 (58.6)	222 (41.3)	<0.001 *
	CRP	>10 mg/L	624 (62.4)	53.32	388 (62.1)	236 (37.8)	<0.001 *
	RBS	>140 mg/dL	496 (49.6)	208.38	326 (65.7)	170 (34.2)	<0.001 *

* Chi square; * *p*-value < 0.05 is statistically significant.

**Table 4 healthcare-11-01192-t004:** Univariable and multivariable analysis on the predictors of death.

Variables	Groups	N (%)	Univariate Logistic Regression OR (95% CI)	*p*-Value	Multivariate Logistic Regression OR (95% CI)	*p*-Value
Age	Above 40	727 (72.7)	1.925 (1.376–2.694)	<0.001 *	1.141 (0.631–2.064)	0.662
Gender	Male	597 (59.7)	1.422 (1.070–1.891)	0.015 *	1.390 (0.857–2.254)	0.181
Co-morbidities	Yes	571 (57.1)	2.089 (1.562–2.794)	<0.001 *	0.528 (0.279–0.998)	0.049 *
On ventilator	Yes	237 (23.7)	86.63 (52.94–141.7)	<0.001 *	124.267 (65.959–234.119)	<0.001 *
Oxygen Sat	Less than 95%	750 (75.0)	2.417 (1.678–3.482)	<0.001 *	3.234 (1.554–6.728)	0.002 *
Tachycardia	Yes	306 (30.6)	1.736 (1.301–2.317)	<0.001 *	1.091 (0.657–1.811)	0.738
SOB	Yes	795 (79.5)	3.559 (2.284–5.547)	<0.001 *	2.595 (1.203–5.597)	0.015 *
High Blood pressure	Yes	399 (39.9)	2.021 (1.531–2.668)	<0.001 *	1.149 (0.657–2.008)	0.626
Hyperglycemia	Yes	496 (49.6)	1.706 (1.293–2.251)	<0.001 *	0.956 (0.566–1.613)	0.865
WBC	>10 × 10^9^/L	691 (69.1)	2.563 (1.830–3.591)	<0.001 *	0.923 (0.514–1.657)	0.789
Lymphocytopenia	<40%	725 (72.5)	1.398 (1.016–1.922)	0.039 *	1.372 (0.784–2.402)	0.268
Thrombocytopenia	Yes	171 (17.1)	1.772 (1.257–2.499)	0.001 *	1.161 (0.624–2.163)	0.637
AST	>40 units/L	437 (43.7)	2.553 (1.928–3.380)	<0.001 *	1.555 (0.962–2.513)	0.071
Bilirubin	>1.2 mg/dL	75 (7.5)	5.074 (3.096–8.315)	<0.001 *	2.409 (1.051–5.522)	0.038 *
Serum Creatinine	>1.2 mg/dL	247 (24.7)	2.359 (1.743–3.191)	<0.001 *	2.344 (1.384–3.968)	0.002 *
D dimer	>0.50	537 (53.7)	4.239 (3.103–5.792)	<0.001 *	1.659 (1.010–2.727)	0.046 *
CRP	>10 mg/L	624 (62.4)	3.790 (2.712–5.295)	<0.001 *	2.979 (1.686–5.261)	<0.001 *
Antibiotics	Given	779 (77.9)	2.772 (1.860–4.132)	<0.001 *	1.316 (0.619–2.798)	0.475
Corticosteroids	Given	548 (54.8)	1.623 (1.226–2.150)	0.001 *	0.644 (0.334–1.242)	0.189
Anticoagulants	Given	340 (34.0)	1.190 (0.894–1.584)	0.234	1.378 (0.794–2.393)	0.254
Antivirals	Given	289 (28.9)	1.722 (1.285–2.306)	<0.001 *	0.475 (0.235–0.959)	0.038 *
Tocilizumab	Given	203 (20.3)	1.944 (1.409–2.683)	<0.001 *	0.896 (0.461–1.745)	0.748
Ivermectin	Given	92 (9.2)	0.809 (0.494–1.324)	0.399	0.518 (0.222–1.211)	0.129

* Chi square; * *p*-value < 0.05 is statistically significant.

## Data Availability

The data supporting the findings of this study are available from the corresponding authors upon reasonable request.

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
