# Peer review of "Clinical Disease Characteristics and Treatment Trajectories Associated with Mortality among COVID-19 Patients in Punjab, Pakistan"

_healthcare, 2023, doi:10.3390/healthcare11081192_

Round 1

Reviewer 1 Report (New Reviewer)

The title needs to be rephrased, I would delete such as mortality and may be replace it in relation to mortality etc, try to make it precise and specific

The Introduction covers all aspects of COIVD-19 including history of the pandemic, the affects on healthcare systems, therapeutic plans and pharmacological options including advantages and disadvantages.

The last paragraph of the introduction should clearly state the purpose of the study. I’d suggest adding a closing statement saying so the aim of the study is …

Methodology

What do you mean by not admitted ? outpatients ?

Was data collection done in different settings for inpatients and outpatients ?

You stated later in outcomes that patients in ICU were investigated, but you previously mentioned patients not admitted.

The outcome measures should be specific

“We analyzed the COVID-19 infection occurrence, intensive care admission or me- 179 chanical ventilation and mortality during the different pandemic waves in Pakistan”

Is this a general statement or your outcome variables?

Results are comprehensive and well supported with tables. Data interpretation is well written, and regression test perfectly used to correlate demographics, pharmacological options, co-morbidities with the primary outcome i.e mortality.

Discussion is supported with relevant literature( 131 studies are included) from other countries. The study strengths and limitations are also included.

Conclusion is not available  

Author Response

Dear Reviewer,

Thank you very much for sending us the valuable comments on our manuscript

“Clinical disease characteristics and treatment trajectories in relation to outcome such as mortality in COVID-19 patients.”. This response letter summarizes our revisions and addresses the points raised by the respected reviewer. These are as follows:

Point 1: The title needs to be rephrased, I would delete such as mortality and may be replace it in relation to mortality etc, try to make it precise and specific.

Response 1: Title has been rephrased as:

“Clinical disease characteristics and treatment trajectories in relation to mortality among COVID-19 patients”

Point 2: The last paragraph of the introduction should clearly state the purpose of the study. I’d suggest adding a closing statement saying so the aim of the study is …

Response 2: Paragraphs highlighting above mentioned missing ponts have been added at the end of the introduction. Further guidnce if provided would be highly appreciated.

Our study's aims to determine the risk factor prevalence and to analyze how these variables affect the COVID-19 disease's outcomes, such as mortality. In this large sample size multicenter study, we examined the relationship between clinical illness characteristics, pharmacological medications provided and COVID-19-related mortality in Pakistan across different phases of the pandemic.

Numerous initiatives have been launched on the same subject. However, the majority of the proof is founded on research done in the early phases of the pandemic. Therefore, a bigger population-based research is urgently needed to update the data. The continued investigation and accumulation of data from each period will also help to shed light on critical lessons for the generations to come. Addition-ally, there is little proof of how COVID-19 impacts vary geographically in low and middle-income nations like Pakistan, where disparities in access to high-quality healthcare services may greatly affect mortality risk.

Furthermore, COVID-19 treatments have evolved over time, and the value of contemporary pharmacological regimens is still up for discussion. In regression models, we compared patients who had received each specific treatment to those who had not in order to investigate therapeutic effects.

For a sizeable portion of patients, COVID-19 is still potentially fatal. This research can assist medical professionals in identifying high risk patients who will require specialized treatment quickly and developing emergency triage strategies. Such patients are more likely to get severely sick or even pass away and need admittance to the ICU, by doing this, more ICU spaces can be spared. Providing patients with risk factors with urgent care quicker can lower fatality rates. These epidemiological studies are essential to support clinical judgment and logistical preparation for enhancing patient treatment and the caliber of healthcare services. As treating a lot of people puts a lot of strain on medical re-sources, this will also alleviate pressure on medical services.

Point 3: What do you mean by not admitted ? outpatients ?

Response 3: Apologies for the misunderstanding, actually we intended to include all patients with confirmed COVID-19 infection, either treated in their homes or admited to a hospital.

But by chance , all the data we collected involves patients who were admitted to hospital for COVID-19 treatment. So non of the pateint included was treated in out-patent setting.

Confusing statement has been re-phrased for clarity.

Point 4: Was data collection done in different settings for inpatients and outpatients ?

Response 4: Data collection was done in in-patient settings in different hospitals. Confusing statement has been re-phrased for clarity.

Point 5: You stated later in outcomes that patients in ICU were investigated, but you previously mentioned patients not admitted.

Response 5: Again, apologies for confusing statement. As stated above, all the patients included in this study were admitted in a hospital for COVID-19 treatement. So information of these patients admitted in different wards of hospital including ICU was also collected and evaluated. Confusing statement has been re-phrased for clarity.

Point 6: The outcome measures should be specific

“We analyzed the COVID-19 infection occurrence, intensive care admission or me- 179 chanical ventilation and mortality during the different pandemic waves in Pakistan”

Is this a general statement or your outcome variables?

Response 6: There was only one specific primary outcome i.e .mortality.

Mortality outcome as a dependent variablee was linked with independent variables such as patients demographics, disease features and pharmacological treatments provided.

Point 7: Results are comprehensive and well supported with tables. Data interpretation is well written, and regression test perfectly used to correlate demographics, pharmacological options, co-morbidities with the primary outcome i.e mortality.

Response 7: Dear Reviewer, we are grateful for encouraging feedback and for giving us the possibility for a revision according to constructive discussion of weaknesses in our paper.

Point 8: Discussion is supported with relevant literature( 131 studies are included) from other countries. The study strengths and limitations are also included.

Response 8: Dear Reviewer, we are grateful for encouraging feedback and for giving us the possibility for a revision according to constructive discussion of weaknesses in our paper.

Point 9: Conclusion is not available

Response 9: Conclusion, study strenghts and limitations have been added in the re-submitted manuscript.

For more details please see the revised version manuscript.

Reviewer 2 Report (New Reviewer)

Dear authors, thank you for conducting this study which could be interesting to clinicians monitoring this topic. Nevertheless, after more than three years since the pandemic started, and the abundance of articles with similar subjects and conduct, I could not notice the novelty of this article. Therefore, I regretfully should advise against its acceptance.

Author Response

Dear Reviewer,

Thank you very much for sending us the valuable comments on our manuscript .

“Clinical disease characteristics and treatment trajectories in relation to outcome such as mortality in COVID-19 patients.”.

This response letter summarizes our revisions and addresses the points raised by the respected reviewer. These are as follows:

Point 1: Dear authors, thank you for conducting this study which could be interesting to clinicians monitoring this topic.

Response 1: Dear Reviewer, we are grateful for encouraging feedback and for giving us the possibility for a revision according to constructive discussion of weaknesses in our paper.

Point 2: Nevertheless, after more than three years since the pandemic started, and the abundance of articles with similar subjects and conduct, I could not notice the novelty of this article. Therefore, I regretfully should advise against its acceptance.

Response 2: We agree with the reviewer that there are the abundance of articles with similar subjects and conduct. However, following are some points we would like to mention as the importance of this study.

  • Numerous initiatives have been launched on the same subject. However, the majority of the proof is founded on research done in the early phases of the pandemic[1-3]. Therefore, a bigger population-based research is urgently needed to update the data[4]. The continued investigation and accumulation of data from each period will also help to shed light on critical lessons for the generations to come. These risk factors are specifically used to determine the treatment plan and priority group for immunization or medical care and are publicized in each country's guidelines [5]. Additionally, there is little proof of how COVID-19 impacts vary geographically in low and middle-income nations like Pakistan, where disparities in access to high-quality healthcare services may greatly affect mortality risk.
  • For a sizeable portion of patients, COVID-19 is still potentially fatal. This research can assist medical professionals in identifying high risk patients who will require specialized treatment quickly and developing emergency triage strategies[6]. Such patients are more likely to get severely sick or even pass away and need admittance to the ICU, by doing this, more ICU spaces can be spared. Providing patients with risk factors with urgent care quicker can lower fatality rates[7, 8]. These epidemiological studies are essential to support clinical judgment and logistical preparation for enhancing patient treatment and the caliber of healthcare services.
  • Furthermore, COVID-19 treatments have evolved over time, and the value of contemporary pharmacological regimens is still up for discussion. In regression models, we compared patients who had received each specific treatment to those who had not in order to investigate therapeutic effects[9].
  • Strengths of our multicentre reports are the relatively large number of patients[10, 11] from two geographically diverse hospitals in Pakistan, as well as the real-world settings. Other strengths are use of simple clinical parameters as tool to predict COVID-19 mortality[9], its applicability in triaging patients at time of admission and use of logistic regression model in order to examine the association among the variables and COVID-19 related mortality[12].
  • There are also evident limitations. In this retrospective analysis, there is no control group. The therapies were administered in a range of doses, modes, and durations, and the severity of COVID-19 disease was not consistent, ranging from patients who did not require oxygen to mechanically ventilated patients in intensive care units[10]. The patients were recruited from a specific region, and our results might not apply in other countries as factors associated with mortality may differ in various regions

References:

  1. Kim, L., et al., Risk factors for intensive care unit admission and in-hospital mortality among hospitalized adults identified through the US coronavirus disease 2019 (COVID-19)-associated hospitalization surveillance network (COVID-NET). Clinical infectious diseases, 2021. 72(9): p. e206-e214.
  2. Parra-Bracamonte, G.M., N. Lopez-Villalobos, and F.E. Parra-Bracamonte, Clinical characteristics and risk factors for mortality of patients with COVID-19 in a large data set from Mexico. Annals of epidemiology, 2020. 52: p. 93-98. e2.
  3. Soares, R.d.C.M., L.R. Mattos, and L.M. Raposo, Risk factors for hospitalization and mortality due to COVID-19 in Espírito Santo State, Brazil. The American journal of tropical medicine and hygiene, 2020. 103(3): p. 1184.
  4. Miyashita, K., et al., Changes in the characteristics and outcomes of COVID-19 patients from the early pandemic to the delta variant epidemic: a nationwide population-based study. Emerging Microbes & Infections, 2023. 12(1): p. 2155250.
  5. Kleinkauf, N., et al., European Centre for Disease Prevention and Control: risk assessment on the impact of environmental usage of triazoles on the development and spread of resistance to medical triazoles in Aspergillus species. ECDC, Solna, Sweden, 2013.
  6. Pijls, B.G., et al., Demographic risk factors for COVID-19 infection, severity, ICU admission and death: a meta-analysis of 59 studies. BMJ open, 2021. 11(1): p. e044640.
  7. Shayganfar, A., et al., Risk factors associated with intensive care unit (ICU) admission and in-hospital death among adults hospitalized with COVID-19: a two-center retrospective observational study in tertiary care hospitals. Emergency Radiology, 2021. 28: p. 691-697.
  8. Dongelmans, D.A., et al., Characteristics and outcome of COVID-19 patients admitted to the ICU: a nationwide cohort study on the comparison between the first and the consecutive upsurges of the second wave of the COVID-19 pandemic in the Netherlands. Annals of intensive care, 2022. 12(1): p. 1-10.
  9. Sharifi, F., et al., Clinical Risk Factors of Need for Intensive Care Unit Admission of COVID-19 Patients; a Cross-sectional Study. Archives of Academic Emergency Medicine, 2023. 11(1): p. e15-e15.
  10. Laldinmawii, G., et al., Epidemiological study of COVID-19 in Mizoram, India: Meta-analysis of sociodemographic determinants, risk factors, and outcome. Asian Journal of Medical Sciences, 2023. 14(1): p. 3-9.
  11. Qureshi, M.A., K.U. Toori, and R.M. Ahmed, Predictors of Mortality in COVID-19 patients: An observational study. Pakistan Journal of Medical Sciences, 2023. 39(1).
  12. Subudhi, S., et al., Comparing machine learning algorithms for predicting ICU admission and mortality in COVID-19. NPJ digital medicine, 2021. 4(1): p. 87.

For more details please see the revised version manuscript.

Reviewer 3 Report (New Reviewer)

The present retrospective study evaluates several clinical and therapeutic parameters that may influence the management of COVID-19 patients. The topic is relevant, but requires modification and detailing of certain aspects to improve the information and structure of the present article.

Shape suggestions

Please revise the template and instructions for authors. A period is not necessary as a punctuation mark after the publication type or title.

Abstract- L30 and L34 - it is not necessary to capitalize "remdesivir" and "patients".

L35- to use the abbreviated form CRP, should be explained at its first appearance in the abstract (L24).

The abstract and main text are two separate sections in terms of the use of abbreviations. Please revise the instructions for authors.

L50 - [7,8] bibliographic clues are per se structures of the article and are not linked to any word in the text. Please revise the whole manuscript from this point of view.

The information in the introduction section is organized in the form of overly long paragraphs, which decreases readability and comprehension. Please reorganize into shorter paragraphs that will be more logical and easier to understand.

L153- not relevant to present the author's position within the journal or the PMID of the article. It is sufficient to present in a proper way the idea supported by the article, especially as the bibliographic resource is already indicated.

No colon is needed for subsections.

L172-"Kidney and liver function status" is not a sentence.

L184- The version of the statistics software should be presented together with details about the country, city.

There is no need for a period (.) in some columns of the tables.

Content suggestions

L44- There are several variants that have been widespread or are currently being monitored, the most prevalent and noteworthy of which is the omicron variant. A comparative presentation of these variants clinically based on virulence is needed.

The purpose of the paper, the motivation for the choice of the topic, the contribution to the field, and the novelty of the study are missing from the last paragraph of the introduction. This needs to be revised.

Exclusion criteria should also be detailed because they also directly influence the final results of the study.

Remdesivir is, however, one of the molecules extensively used in COVID-19, and a number of studies have shown favorable results. Since it is a widely used repurposing drug, it is necessary to detail the pharmacological mechanisms of RDV in order to comprehensively describe the interaction between the viral particle and the drug, from which conclusions can also be drawn about the mechanisms of resistance. I suggest checking and referring to: PMID: 35131656.

The clinical relevance of biomarkers used in the management of COVID-19 has been discussed to a limited extent. Adiponectin has been intensively studied in this regard and is worth mentioning for the other biomarkers with implications in COVID-19. I suggest checking and referring to: PMID: 36211634

Given that data on the comorbidities of infected patients have been obtained, it is important to describe the most common and clinically relevant comorbidities/associated diseases in order to create a framework for describing the extent to which they may influence therapeutic outcomes. I suggest you check and refer to: PMID: 36406478

A conclusion section is needed to summarize the most important aspects of the publication associated with future research directions that can address the limitations of the current study.

Author Response

Dear Reviewer,

Thank you very much for sending us the valuable comments on our manuscript

“Clinical disease characteristics and treatment trajectories in relation to outcome such as mortality in COVID-19 patients.”. Additionally, we are grateful for encouraging feedback and the decision for giving us the possibility for a revision according to constructive discussion of weaknesses in our paper. This response letter summarizes our revisions and addresses the points raised by the respected reviewer.These are as follows:

Point 1: Abstract- L30 and L34 - it is not necessary to capitalize "remdesivir" and "patients"..

Response 1: We are sorry for this mistake, correction has been made and included in re-submitted manuscript file.

Point 2: L35- to use the abbreviated form CRP, should be explained at its first appearance in the abstract (L24)

Response 2: Again, We are sorry for this mistake, correction has been made and included in re-submitted manuscript file.

Point 3: L50 - [7,8] bibliographic clues are per se structures of the article and are not linked to any word in the text. Please revise the whole manuscript from this point of view.

Response 3: This was done as per the submission instructions, however all the bibliographic clues in the article are now linked to a relevant statement/word in the text as per suggested.

Point 4: The information in the introduction section is organized in the form of overly long paragraphs, which decreases readability and comprehension. Please reorganize into shorter paragraphs that will be more logical and easier to understand."..

Response 4: . The information in the introduction section has been reorganized into shorter paragraphs that would be easier to understand, further guidance if provided would be highly appreciated.

Point 65 L153- not relevant to present the author's position within the journal or the PMID of the article. It is sufficient to present in a proper way the idea supported by the article, especially as the bibliographic resource is already indicated.

Response 5: Authors position and the PMID has been removed. Idea supported by the article as the bibliographic resource is already indicated. further guidance if provided would be highly appreciated.

Point 6: No colon is needed for subsections.

Response 6: Correction has been made and included in re-submitted manuscript file.

Point 7: L172-"Kidney and liver function status" is not a sentence.

Response 7: Correction has been made and included in re-submitted manuscript file.

Point 8: L184- The version of the statistics software should be presented together with details about the country, city.

Response 8: The version of the statistics software SPSS is now presented together with details about the country, city in the re-submitted manuscript.

Point 9: There is no need for a period (.) in some columns of the tables.

Response 9: Correction has been made and included in re-submitted manuscript file.

Point 10 L44- There are several variants that have been widespread or are currently being monitored, the most prevalent and noteworthy of which is the omicron variant. A comparative presentation of these variants clinically based on virulence is needed.

Response 10: A diagram with a comparative presentation of COVIID-19 virus variants based on virulence is added.

Point 11 The purpose of the paper, the motivation for the choice of the topic, the contribution to the field, and the novelty of the study are missing from the last paragraph of the introduction. This needs to be revised.

Response 11: Paragraphs highlighting above mentioned missing ponts have been added at the end of the introduction.Further guidnce if provided would be highly appreciated.

Our study's aims to determine the risk factor prevalence and to analyze how these variables affect the COVID-19 disease's outcomes, such as mortality. In this large sample size multicenter study, we examined the relationship between clinical illness characteris-tics, pharmacological medications provided and COVID-19-related mortality in Pakistan across different phases of the pandemic.

Numerous initiatives have been launched on the same subject. However, the majority of the proof is founded on research done in the early phases of the pandemic. Therefore, a bigger population-based research is urgently needed to update the data. The continued investigation and accumulation of data from each pe-riod will also help to shed light on critical lessons for the generations to come. Addition-ally, there is little proof of how COVID-19 impacts vary geographically in low- and mid-dle-income nations like Pakistan, where disparities in access to high-quality healthcare services may greatly affect mortality risk.

Furthermore, COVID-19 treatments have evolved over time, and the value of contemporary pharmacological regimens is still up for discus-sion. In regression models, we compared patients who had received each specific treat-ment to those who had not in order to investigate therapeutic effects.

For a sizeable portion of patients, COVID-19 is still potentially fatal. This research can assist medical professionals in identifying high risk patients who will require specialized treatment quickly and developing emergency triage strategies. Such patients are more likely to get severely sick or even pass away and need admittance to the ICU, by doing this, more ICU spaces can be spared. Providing patients with risk factors with urgent care quicker can lower fatality rates. These epidemiological studies are essential to support clinical judgment and logistical preparation for enhancing patient treatment and the cali-ber of healthcare services. As treating a lot of people puts a lot of strain on medical re-sources, this will also alleviate pressure on medical services.

Point 12 Exclusion criteria should also be detailed because they also directly influence the final results of the study.

Response 12:Inclusion and Exclusion criteria has been detailed in the first subsection of Materials and Methods.

Point 13: Remdesivir is, however, one of the molecules extensively used in COVID-19, and a number of studies have shown favorable results. Since it is a widely used repurposing drug, it is necessary to detail the pharmacological mechanisms of RDV in order to comprehensively describe the interaction between the viral particle and the drug, from which conclusions can also be drawn about the mechanisms of resistance. I suggest checking and referring to: PMID: 35131656.

Response 13: As suggested we have referred to : PMID: 35131656 and some other studies with details regarding mechanism of action of Remdesivir and added them in introduction.

Point 14: The clinical relevance of biomarkers used in the management of COVID-19 has been discussed to a limited extent. Adiponectin has been intensively studied in this regard and is worth mentioning for the other biomarkers with implications in COVID-19. I suggest checking and referring to: PMID: 36211634

Response 14: As suggested we have referred to : PMID: 36211634 and some other studies with details on biomarkers such as D-dimers, CRP and Adiponectin and added relevant information in introduction.

Point 15: Given that data on the comorbidities of infected patients have been obtained, it is important to describe the most common and clinically relevant comorbidities/associated diseases in order to create a framework for describing the extent to which they may influence therapeutic outcomes. I suggest you check and refer to: PMID: 36406478

Response 15: As suggested we have referred to : PMID: 36406478 and some other studies with details on co-morbidities in order to create a framework for describing the extent to which they may influence therapeutic outcome and added relevant information in introduction.

Point 16: conclusion section is needed to summarize the most important aspects of the publication associated with future research directions that can address the limitations of the current study.

Response 16: Conclusion, study strenghts and limitations have been added in the re-submitted manuscript.Which is as follows:

The COVID-19 epidemic has had a significant effect on social, economic, and health conditions, pushing healthcare systems to their breaking point all over the world. According to this research, there were recurring risk factors for rising mortality across the pandemic phases. Age, male sex, and pre-existing comorbidities are revealed to be demographic risk factors for the onset of severe illness and even mortality, as seen in the multivariate analysis. The intensity and mortality of COVID-19 may also be predicted by laboratory markers that point to problems during hospitalization, such as the onset of acute respiratory distress syndrome or failure of the organs such as liver and kidneys.

Therefore, COVID-19 patients who have mortality indicators should be quickly admitted to the hospital, evaluated, and treated before their clinical state deteriorate. In order to protect global health from the current COVID-19 pandemic and future dangers, it is essential to guarantee health equity in less developed nations with subpar health services. to develop systems that allow for early access to healthcare, which are essential for lowering the mortality rate from severe COVID-19, especially in low and middle-income countries.

Strengths of our reports are the relatively large number of patients from two geographically diverse hospitals in Pakistan, as well as the real-world settings. Other strengths are use of simple clinical parameters as tool to predict COVID-19 mortality, its applicability in triaging patients at time of admission and use of logistic regression model in order to examine the association among the variables and COVID-19 related mortality.

There are also evident limitations. In this retrospective analysis, there is no control group. The therapies were administered in a range of doses, modes, and durations, and the severity of COVID-19 disease was not consistent, ranging from patients who did not require oxygen to mechanically ventilated patients in intensive care units.

For more details please see the revised version manuscript.

Round 2

Reviewer 3 Report (New Reviewer)

Please see my minor requests: 

Abstract remained much too long. Please revise it as numbers of characters, as the Instructions for authors requests.

L184-209. Aim of the study, I requested the novelty/special aspects your study brings to the field, NOT describing what you have done during the research you have performed (and can be seen in the manuscript, in the other sections). Moreover, AIM of the study should have NO reference, as it is YOUR own. Please correct.

What about Figure 1.? Before inserting any Figure or Table it must be mentioned in the main text (Figure 1 depicts, Figure 2 presents, Figure 3 describes… Table 2 summarizes…). Please check and complete the entire manuscript. Also, data mentioned there must be referenced, as it is part of Introduction (background). I suggest checking and referring to PMID: 34863742 and https://doi.org/10.1016/j.biopha.2022.112756

Also, Figure 1 is blurred. Please replace it with a best quality one. Do not save it in any format, only print screen it from its original version.

References should be written in the MDPI style, please check the Instructions for authors.

Author Response

This manuscript is a resubmission of an earlier submission. The following is a list of the peer review reports and author responses from that submission.

Round 1

Reviewer 1 Report

I was very interested in reviewing this article considering the clinical long-term consequences of COVID-19 pandemic, very frequently encountered in our daily practice.

So, the epidemiological value of the study is important for completing the actual data and the global view about this pandemic. The results are coherent and adequately supported by statistical analysis. I also appreciate the results are synthetized in the four tables.

On the other hand, the novelty of scientific information is poor, only sustained by well-known pathogenic mechanisms and the similarities with other studies.

I suggest the authors to complete the discussion by comparing the actual results with other Asian studies, considering the importance of genetic characteristics, socio-economic and behavioral regional factors.

COVID acronym includes the term of infectious disease. Please correct “COVID infection” – rows 58 and 60, 91.

Author Response

Dear Reviewer:

"Please see the attachment".

Reviewer 2 Report

P-value > 0.05 was used to identify 104 statistically significant observations throughout the statistical analysis. Please, explain the meaning of this sentence.

The abbreviation “bpm” should be explained. It probably denotes different units (heart rate and respiration, respectively).

The authors wrote on Ivermectin and supported their postulate by reference [10].

However, “The Editor of the American Journal of Therapeutics hereby issues an Expression of Concern for Bryant A, Lawrie TA, Dowswell T, Fordham EJ, Mitchell S, Hill SR, Tham TC. Ivermectin for Prevention and Treatment of COVID-19 Infection: A Systematic Review, Meta-analysis, and Trial Sequential Analysis to Inform Clinical Guidelines. Am J Ther. 2021;28(4): e434-e460.” .

This concern (PMID: 35142702) must be mentioned.

Statements such as: “Viral infection reduces the production of interleukin (IL)-17, which is crucial for the clearance of bacteria by neutrophils. In addition, influenza virus infections promote S. aureus colonization.”, should be supported by references.

The authors wrote: “In univariate analysis of medications given, we found no evidence that apart from 277 the use of ivermectin, other drugs, such as antibiotics, corticosteroids, anticoagulant anti-278 virals and even tocilizumab, prevent mortality in patients with COVID-19. However, mul-279 tivariate analysis showed the opposite results for corticosteroids, antiviral drugs and to-280 cilizumab. Our findings are consistent with those of other studies[33].” Not only that it is difficult to understand the meaning of these sentences, reference [33] is entitled: “Human kidney is a target for novel severe acute respiratory syndrome coronavirus 2 infection”.

The authors wrote: “International remdesivir trials in Japan also revealed that an antiviral medicine reduces the probability of hospitalization[36].”

In that reference [36] (based on patients admitted to a large community hospital in Northeast Georgia) I found the only the following on remdesivir: “Patients who received high-dose corticosteroids more often received other possible therapies for COVID-19 (tocilizumab, remdesivir, and higher dose anticoagulation).”.

Reference 37 cites tocilizumab in the Recovery trial, which is a separate publication.

In brief, the Reference list does not correspond to the references in the text.

However, even if the Reference list section is corrected, I cannot see that this study adds any new information to the knowledge paradigm on COVID-19.

Round 2

Reviewer 1 Report

same as for editor

Author Response

Highly grateful to the respected reviewer for providing valuable input which helped in further improvement of the manuscript.

Reviewer 2 Report

The authors have explained the abbreviations used and made some slight revisions of the text, e.g. p. 2 Lines 71-75.

However, the scientific value of the manuscript is limited.
Several reviews addressing the topic described by the authors, already exist; e.g. PMID: 34418980.

Author Response

Highly grateful to the respected reviewer for providing valuable input and guidance which really helped in further improvement of the manuscript.